# Changes in the Physical–Chemical Properties and Volatile Flavor Components of Dry-Cured Donkey Leg during Processing

**DOI:** 10.3390/foods11213542

**Published:** 2022-11-07

**Authors:** Jingjing Zhang, Zixiang Wei, Huachen Zhang, Lan Xie, Silvia Vincenzetti, Paolo Polidori, Lanjie Li, Guiqin Liu

**Affiliations:** 1Shandong Engineering Technology Research Center for Efficient Breeding and Ecological Feeding of Black Donkey, School of Agriculture Science and Engineering, Liaocheng University, Liaocheng 252000, China; 2School of Biosciences and Veterinary Medicine, University of Camerino, Via Circonvallazione 93, 62024 Matelica, Italy; 3School of Pharmacy, University of Camerino, Via Gentile da Varano, 62032 Camerino, Italy

**Keywords:** donkey, hind leg, chemical composition, free amino acids, free fatty acids, volatile compounds

## Abstract

In order to explore the quality variation and flavor formation of dry-cured donkey leg, the changes in physical–chemical composition, lipolytic, free amino acids content and volatile flavor compounds were investigated in this study. Six fresh, trimmed hind legs with average weight of 8.12 ± 0.8 kg were taken from male Dezhou donkeys slaughtered at the age of 24 months with the average live weight of 240 kg. The entire processing time was eight months long including six stages, specifically: cooling, salting, air-drying, fermenting and aging. Samples were collected at 0 d, 10 d, 20 d, 30 d, 65 d, 105 d and 165 d of processing. The results showed that the pH value remained stable in the range of 6.2~6.6. The moisture and water activity significantly decreased (*p* < 0.05) during processing. The chloride content, ash, total volatile basic nitrogen (TVB-N) and peroxide value (POV) significantly increased (*p* < 0.05), from 0.45% to 12.39%, from 3% to 17%, from 1.43 mg/kg to 8.98 mg/kg and from 1.39 g/100 g to 5.26 g/100 g, respectively. The thiobarbituric acid (TBARS) value reached its highest value of 0.39 mg MDA/kg at the end of the salting stage and then decreased to 0.34 mg MDA/kg. Eighteen free amino acids and fifteen free fatty acids were detected, and their contents were significantly increased during processing (*p* < 0.05). Volatile compounds were analyzed using solid-phase microextraction (SPME) and gas chromatography–mass spectrometry (GC–MS). Among 114 volatile compounds detected in dry-cured donkey leg, aldehydes, esters, alkane and alcohols were more abundant in the final products, with relative concentrations of 41.88%, 5.72%, 5.35% and 5.25%, respectively. Processing significantly affected the physical–chemical properties, which could contribute to the formation of flavor substances of dry-cured donkey leg.

## 1. Introduction

Dry-cured ham has been considered a very popular meat product all over the world, due to its bright color, moderate texture and pleasant characteristic aroma. Among them, Chinese dry-cured hams including Jinhua ham, Xuanwei ham and Rugao ham, are mainly produced in Yunnan and Zhejiang provinces [1,2,3]. In addition, other internationally famous dry-cured hams, such as Spain’s Iberian ham [4], Italy’s Parma ham [5] and France’s Bayonne ham [6], have been also deeply investigated. As far as we know, a large number of dry-cured hams marketed all over the world are processed using pork legs [5,7], and a few are made from mutton in recent years [8]. Donkey meat is required by an increasing number of consumers in different areas of the world due to its tender texture, peculiar taste and crucial nutritional compounds [9,10]. In fact, donkey meat shows a high content of protein, essential amino acids and essential fatty acids, while fat content, cholesterol content and energy supply are low [11,12]. Today, China is the top producer of donkey meat in the world, followed by Niger [11]. Dry-cured donkey leg can be a new potential functional food with high nutritional value. Several previous studies have well-reported the changes in quality of pork and mutton ham during processing [13,14,15]; however, no information on the ham prepared using donkey meat were reported until now. Therefore, it is very important to investigate how physical and chemical properties and volatile flavor compounds of dry-cured donkey leg are affected during processing to develop better processing strategies for donkey leg.

Generally, dry-cured ham has a long processing period, usually ranging 8–24 months [16]. During processing, the major biochemical reactions including proteolysis, lipid oxidation, Maillard reactions and Strecker degradation are essential factors affecting the development of the typical aroma of dry-cured ham [17,18,19,20]. In Istrian dry-cured ham, 92 volatile aroma compounds have been identified [21], which belong to several chemical groups: 27 terpenes, 23 aldehydes, 14 alcohols, 10 ketones, 6 alkanes, 4 esters, 6 aromatic hydrocarbons and 2 acids. More than 407 volatile flavor substances were found in the Dahe black pig ham, and the most abundantly detected flavor families were aldehydes and alcohols, which derive from oxidation of fatty acids and degradation of amino acids [22]. In Celta dry-cured ham, 22 amino acids were detected [23] through the manufacturing process, and the total content of free amino acids (FAAs) increased significantly (*p* < 0.001). The concentration of FAAs in dry-cured meat products can directly increase flavor intensity [24] and also serve as substrates for aroma volatiles through the Maillard reaction and Strecker degradation with other substances [25]. In addition, FAAs are considered as the quantitative indicator of proteolytic activity [24]. Proteolysis occurs mainly by two proteases, endopeptidases (e.g., calpains and cathepsins) and exopeptidases (e.g., peptidase and aminopeptidase) [26,27]. Lipolysis and lipid oxidation that occurred through processing of dry-cured ham played an important role in the final sensory quality [28]. Free fatty acids (FFAs) derived from lipolysis were the major precursors of volatile compounds, while the volatiles generated by lipid oxidation usually contributed to the aroma characteristic of dry-cured ham [29,30]. The major enzymes involved in these reactions are muscle lipases and lipoxygenase [31]. Yang et al. [32] discovered that the total contents of free fatty acids in marinated pork meat were decreased after high-pressure treatment, while the levels of volatile compounds from lipid oxidation increased. This result was probably attributed to the decline of lipase activity and the increase in lipoxygenase activity. Aldehydes are generally considered the most abundant flavor compounds of different types of dry-cured hams and are derived from free fatty acids oxidation [33,34]. Other major flavor compounds such as alcohols and ketones are produced from β-ketonic acid decarboxylation or fatty acid β-oxidation; however, esters are products of the esterification of alcohols and carboxylic acids [35,36].

In the present study, physical and chemical parameters such as moisture, water activity, pH, chloride content, ash content, free fatty acids, free amino acids and volatile flavor compounds were investigated during processing of dry-cured donkey leg.

## 2. Materials and Methods

### 2.1. Materials and Manufacturing Process of Dry-Cured Donkey Leg

Six fresh trimmed hind legs with average weight of 8.12 ± 0.8 kg taken from male Dezhou donkeys supplied by Wanshixing Breeding Co., Ltd. (Liaocheng, China) were used in this study. Ten donkeys were slaughtered at an age of ∼2 years with a final similar body weight (240 ± 25 kg). They were stunned by an electrical stunner (about 280 V) and then were slaughtered at Dong-E-E-Jiao Co., Ltd. (Liaocheng, China). These donkeys were reared under the same feeding and living conditions. All experimental procedures were conducted according to the National Institute of Health guidelines for the care and use of laboratory animals (NIH Publications No. 8023) and approved by the Animal Care and Use Committee of Liaocheng University (Liaocheng, China).

The processing of dry-cured donkey leg refers to the method of Jinhua ham, and some aspects have been improved [16]. Firstly, the hind legs from the slaughtered donkeys were cooled down for 36 h (4 °C), then trimmed and shaped. The reshaped legs were salted with the amount of salt equal to 8% of the weight of the legs. Salt was spread over the surface of the hams on the first day. On the second day, 70~80% salts were used to spread on the cut surface of the legs (skin side down). On the third day, salt would be added to the exposed parts. After salting, the salt on the surface of the legs was washed away, and then the legs were hung to dry and ferment.

The temperature and relative humidity (RH) changes were recorded in the processing environment every day during processing: salting period (0~35 d, 7 °C, RH 65%), hang-drying period (35~60 d, 10 °C, RH 74%), early fermentation period (60~95 d, 20 °C, RH 74%), late fermentation period (95~125 d, 27 °C, RH78%), aging period (125~185 d, 27 °C, RH 78%). The whole processing time of dry-cured donkey leg is eight months.

### 2.2. Sampling of Dry-Cured Donkey Leg

Three legs were randomly selected for sampling (2 cm subcutaneously, cubic 20 g) at each key processing stage. That is, fresh leg (0 day), begin of salting (10 days), middle of salting (20 days), end of salting (30 days), begin of fermenting (65 days), end of fermenting (105 days), and end of aging (165 days). The samples were immediately placed in 50 mL tubes (sterile enzyme-free) then stored in a refrigerator (BCD-208K/A NCJN, Shanghai, China) at −80 °C until further analysis. Samples were analyzed in triplicate.

### 2.3. Physicochemical Analysis

Moisture content was determined according to the China National Standard GB 5009.3-2016. A donkey meat sample (3 g) was added to 5 mL of ethanol (96% *v*/*v*) and then was dried in a drying oven (Raypa DO-150, Barcelona, Spain) for 24 h at 103 ± 2 °C. Total chloride was detected according to the method III (argentometry) of the China National Standard GB 5009.44-2016.

The POV value was measured according to the China National Standard GB 5009.227-2016. Briefly, a donkey meat sample (8 g) was homogenized in petroleum ether (40 mL) and extracted for 24 h. The solvent was filtered and removed by rotary evaporation. An Erlenmeyer flask of the mixture solution including 3 g filtrate, 30 mL 60% (*v*/*v*) glacial acetic acid in chloroform and 1 mL KI solution was placed in the dark for 3 min after shaking for 30 s. Then 100 mL distilled water was added in the mixture solution, and the solution was titrated with 0.01 M Na_2_S_2_O_3_ containing starch-iodide indicator to an equivalence point. A reagent blank was prepared in the same way without donkey meat.

The TVB-N value was determined using the second method (Kjeldahl method of nitrogen determination) of the China National Standard GB 5009.228-2016. The distillation operation was performed using an automated Kjeldahl system K 9840.

The ash content was assessed according to the China National Standard GB 5009.4-2016. Donkey meat (3–5 g) was added to 1 mL of magnesium acetate (15% *w*/*v*) in crucibles and then was subjected to 550 ± 25 °C for 5–6 h in a muffle furnace (Vulcan BOX Furnace Model 3-550, Yucaipa, CA, USA).

The pH value was analyzed using a digital pH meter (PHS-25, Shenzhen, China) equipped with a penetration probe, according to the method described in [37]. The 2-thiobarbituric acid reactive substances (TBARS) of the samples were measured according to the method described in [38]. In brief, a donkey meat sample (1 g) was dispersed in 5% trichloroacetic acid (5 mL) and then was homogenized for 3 min using an Ultra-Turrax (Ika T25 basic, Staufen, Germany). The homogenate was centrifuged (2050× *g*, 15 min) and remained in the supernatant. The supernatant was filtered, and 5 mL of it was reacted with 5 mL TBA solution (0.02 M). After incubating in a water bath at 96 °C for 40 min, the mixture was measured for absorbance at a wavelength of 532 nm. The TBARS values were calculated from a standard curve and expressed as milligrams of malondialdehyde (MDA)/kg sample.

### 2.4. Free Amino Acid Analysis

#### 2.4.1. Sample Pretreatment

Free amino acids (FAAs) were extracted and determined according to the previously method reported by Lorenzo et al. [39]. For determining FAAs, a freeze-dried meat sample (1 g) was taken using the electronic scale and put in a cold mortar. After grinding with 1 mL 0.02 N HCl, the solution was transferred to a 10 mL centrifuge tube and then processed until constant volume was reached at 5 mL. The mixture was shaken for 10 min at 4 °C and was placed in the refrigerator (4 °C) for 30 min or more. Then the mixture was centrifuged for 15 min (4 °C, 1800× *g*), and the precipitation was discarded. Next, 1 mL of n-hexane was added in 0.5 mL of supernatant, and the solution was fully shaken, mixed and centrifuged for 10 min (4 °C, 11,100× *g*). The supernatant after centrifugation was discarded, and the underlying solution was stored at 4 °C for further analysis.

#### 2.4.2. Determination of Free Amino Acids

For determining free amino acids (FAA), 0.5 mL 8% sulfosalicylic acid was added into the underlying liquid described before, and the mixture was shaken vigorously for 10 min; then the solution was put on ice for 10 min. After that, the mixed solution was centrifuged for 10 min (4 °C, 1600 r/min). The supernatant was transferred to a new tube and vacuumed to dry at 35–40 °C. Finally, the pH 2.2 buffer solutions was added to the tube for redissolution, diluted twice and filtered with 0.22 μm filter membrane.

The FAAs were determined by the automatic amino acid analyzer (S-433D, Sykam, Germany) equipped with vacuum tube concentrator (TVE-1100, Eyela, Japan). The column used was LCA K07/Li type, and the injection volume was 50 μL. The derived FAAs were monitored at 570 nm and 440 nm, column temperature 38–74 °C and flow rate 0.25 mL/min.

### 2.5. Free Fatty Acid Analysis

#### 2.5.1. Fat Extraction

The lipids in the leg were extracted according to the method described by Folch et al. [40]. To extract fat from meat, the leg samples (5 g) were cut into small pieces and placed into a 50 mL tube, then mixed with 30 mL 2:1 chloroform–methanol mixture (*v*/*v*). The mixture was homogenized at low temperature of 4 °C (4000× *g*, 30 min × 3 times) to a final dilution 80 mL by adding the solution of 2:1 chloroform–methanol mixture (*v*/*v*). After leaving at room temperature for 3 h, it was filtered, and then 0.2 of the volume of the mixed solution (7.3 g/L NaCl, 0.5 g/L CaCl_2_) was added to centrifuge for 15 min (1000× *g*). The lower layer liquid of the mixture after centrifugation was evaporated to dry in a vacuum rotary evaporator (45 °C), and then the sample was stored at −20 °C for further analysis.

#### 2.5.2. Elution of Free Fatty Acids

According to the method described by Garcia Regueiro et al. [41], an aliquot with 20 mg of extracted fat was dissolved by 1 mL chloroform. Then 0.5 mL of the solution was applied to a 200 mg amino-propylsilica minicolumn. After that, 4 mL chloroform-2-propanol alcohol (2:1, *v*/*v*) was used firstly to wash out the neutral lipids, and the free fatty acids was eluted by 6 mL 2% acetic acid in diethyl ethers. Both of the neutral lipids and free fatty acids were collected and stored at −20 °C for further analysis.

#### 2.5.3. Methyl Esterification of Free Fatty Acid

After evaporating the eluted free fatty acid solvent in the tube with a Pressure Blowing Concentrator, 2 mL of boron trifluoride-methanol (14%, m/m) and a few drops of 2.2 dimethoxypropane were added; then we left the solution in water (60 °C, 30 min) to methyl the fatty acids. After cooling, 1 mL water and 1 mL n-hexane were added into the tube, respectively. The upper organic layer was completely collected after vibrating and static stratification. Then 0.1 mL of methyl hepadecanoate n-hexane solution with a concentration of 400 µg/mL was added into the upper organic layer as internal standard. Finally, the solvent was dried with a nitrogen blower, and then the volume was fixed to 0.4 mL with n-hexane for gas chromatographic determination.

#### 2.5.4. Determination of Free Fatty Acids

The free fatty acids analysis was carried out using a GC-7890B GC-MS instrument (Agilent Technologies, Santa Clara, CA, USA) equipped with a TR-FAME GC capillary column (Thermo; 100 m × 0.25 mm, ID × 0.2 µm). The temperature of the sample injector was 280 °C, and the temperature of the flame ionization detector (FID) was 260 °C. The oven temperature profile was as follows: 4 °C/min from 140 °C to 240 °C and held for 5 min. Hydrogen 60 kPa, air 50 kPa, carrier gas (high purity nitrogen) 80 kPa, 20 mL/min; injection volume: 1 uL, split ratio 1:40. The relative content of various free fatty acids was calculated by comparing each peak area of fatty acid methyl ester and methyl heptadecanoate.

### 2.6. Volatile Flavor Compounds Analysis

#### 2.6.1. Extraction of Flavor Compounds

The isolation of volatile compounds was carried out by headspace-solid phase microextraction (SPME). The samples in a refrigerator at −80 °C were taken and thawed at 4 °C for about 30 min. After thawing, they were quickly cut into small particles of 2 mm in size at room temperature, and then 5 g of the small pieces was weighed and put into a 20 mL SPME vial. After that, octanolactone (250 µg) was added into the vial as internal standard, and an SPME fiber was inserted into headspace of the vial. Extraction was carried out at 60 °C for 40 min with stirring in a water bath.

#### 2.6.2. Chromatographic Conditions

The extracted volatile substances were quantified by a 7890A-5975C GC-MS system (Agilent Technologies, Santa Clara, CA, USA). The GC analysis system was as follows: High purity helium was the carrier gas, and the flow rate was 1.2 mL/min. Nonsplit injection mode was adopted, flow rate was 2 mL/min for 2 min and the split ratio was 10:1. The initial temperature was 50 °C and kept for 1 min, 5 °C/min from 50 °C to 100 °C, then 4 °C/min to 140 °C and held for 4 min; after that, 4 °C/min to 180 °C and retained for 2 min, and finally 10 °C/min to 245 °C and held for 3 min.

The mass spectra were obtained using a mass selective detector by electronic impact at 70 eV; quadrupole temperature, 150 °C; ion source temperature, 230 °C; and scan range, 35~550 m/z. The flavor compounds were identified by comparing their mass spectra with those contained in the MAINLIB, NISTDEMO, REPLIB and WILLEY libraries. The matching degree was greater than 80% as the qualitative basis and the flavor compounds were quantified with peak area.

### 2.7. Statistical Analysis

Data obtained from different processing stages were analyzed by one-way analysis of variance (ANOVA) using SPSS Version 19.0 (SPSS Ins., Chicago, IL, USA) and were expressed as means ± standard deviation (SEM, *n* = 6). The mean differences between processing periods were detected with Duncan’s multiple range tests, and the statistical significance was set at *p* < 0.05. The figures were plotted using Origin (version 2020, Hampton, MA, USA).

## 3. Results and Discussion

### 3.1. Changes in Physicochemical Parameters

The changes in the physical and chemical parameters of dry-cured donkey leg during the different processing stages are shown in Table 1. During processing, the moisture content of dry-cured donkey leg was reduced from 72% to 35% with a decrease of 37%, and the highest decrease occurred at the middle of the salting stage (*p* < 0.05). The results may be due to the increased temperature and also due to the effects of salting and drying the legs several times. The moisture content at the aging period (35%) was lower than that one reported by Giovanelli et al. [42], who found the moisture content decreased from 71.12% of the green pig hams and stabilized at 54.0% during the aging period. Zhao et al. [43] and Zhang et al. [44] also found a similar decrease trend of moisture content in Jinhua and Xuanwei hams. The change of water activity was not obvious before the middle of the salting stage (*p* > 0.05) but decreased significantly after the end of the salting stage (*p* < 0.05). Water activity reached 0.78 at the aging stage, which was 0.12 lower than that of the green leg (*p* < 0.05). The decrease in moisture content and water activity was advantageous to the long-term preservation of dry-cured donkey leg. Regarding the pH value, it decreased first and then increased during the whole process but remained stable in the range of 6.24~6.60. At weakly acidic pH, sodium glutamate produced by glutamic acid had the strongest umami taste, which plays an important role in the formation of dry-cured ham flavor [45]. From the end of the salting stage to the aging stage, the pH value suffered small but significant increases from 6.24 to 6.60 (*p* < 0.05). This is due to the increased ambient temperature, which makes the activity of endogenous protease increase and promotes the hydrolysis of proteins to produce amines, ammonia, amino acids and other basic substances [46]. The final pH value (6.60) of dry-cured donkey leg was slightly higher than that of other varieties such as Iberian [47], Serrano [48] and dry-cured hams that matured up to 26 months under “bodega” conditions [49].

As shown in Table 1, the chloride content showed an increasing trend from 0.44% to 12.39%, and the highest increase occurred at the end of the salting stage, which was increased by 3.41% (*p* < 0.05). During the salting stage, NaCl can gradually penetrate from the surface to the inside of dry-cured donkey leg as the diffusion and evaporation of the moisture continue, resulting in an increased level of the chloride content. At the aging stage, chloride content (12.39%) was higher than the values of Iberian (9.29%) and Serrano hams (11.4%); however, they all showed a similar increased trend [50]. Regarding ash content, it increased significantly (*p* < 0.05) during the salting stage (from 3% to 13%) due to the increasing salts content and the loss of moisture in dry-cured donkey legs at this stage. In this research, ash content was held steady at 16% after the salting stage, which was about two times higher than that of Istrian dry-cured ham (8.37%) reported by [51]. Thus, the tested dry-cured donkey leg products resemble much more a “charqui” (salted and dried meat) than a dry-cured ham, which would probably require desalting before eating due to the high level of salt (ash). The TBARS value of dry-cured donkey leg increased quickly during processing, from 0.03 mg MDA/kg of the green leg to the highest value of 0.39 mg MDA/kg at the end of the salting stage. This could be related to that the relatively high temperatures in the early stage of fermentation, promoting the hydrolysis of lipid to produce unsaturated fatty acids, which oxidized into the large amounts of malondialdehyde [52]. After the salting stage, TBARS value showed a slowly decreasing trend and reached 0.34 mg MDA/kg at the aging stage. This was caused by two reasons: on the one hand, as the temperature rose after the salting stage, some endogenous proteins were hydrolyzed to produce endogenous substances with antioxidant function such as free amino acids and small molecule peptides; on the other hand, malondialdehyde were further oxidized to produce small molecule substances such as alcohols and carboxylic acids, resulting in the decrease in malondialdehyde content [53,54]. Some authors found a marked increase (*p* < 0.05) in TBARS value of Iberian ham occurred at the drying stage, which was consistent with the present work [52]. The TBARS value of Duroc dry-cured ham after dry-curing for 12 months detected by Cilla et al. [55] was 0.33 mg MDA/kg, which was in accordance with the value (0.34 mg MDA/kg) obtained in this study. Peroxide value (POV) is an important index reflecting the oxidation degree of fat in meat products. The POV value of dry-cured donkey leg increased significantly (*p* < 0.05) from 1.39 g/100 g at the green leg stage to 5.12 g/100 g at the early fermentation stage, which was due to the rapid accumulation of primary products produced by continuous lipid oxidation. Then the POV value gradually increased to the highest value of 5.26 g/100 g at the aging stage; however, there was no significant difference in that from the early fermentation stage to the aging stage. This was related to some of the primary oxidized products that could be further oxidized to produce alcohol, aldehyde, ketone, acid and other flavor substances after the early fermentation stage. The highest POV value of Laowo ham reported by Wu et al. [14] was 0.13 g/100 g at the early aging stage, which was much less than that of dry-cured donkey leg. The TVB-N value can be used to indicate the degree of meat spoilage or reflect the protein changes during processing stages of cured meat products to evaluate their safety and flavor [56]. The TVB-N content of dry-cured donkey leg increased significantly (*p* > 0.05) throughout the process, with the highest value (8.58 mg/kg) occurring at the aging stage, which was lower than that of the national standard (15 mg/100 g). This can probably be explained by the breakdown of protein producing more ammonia, amines and other basic nitrogen-containing substances under the action of increased protease activity and some bacteria [51]. Evolution of TVB-N content in dry-cured donkey leg was in agreement with Irene Cilla et al. [49], who observed it increased with time and had a good relationship with proteolysis.

### 3.2. Changes in Free Amino Acids

As shown in Table 2, about 18 free amino acids were found in dry-cured donkey leg during processing, including 8 essential amino acids, 2 semiessential amino acids for newborns and 8 nonessential amino acids. Glutamic acid was the most abundant free amino acids at each processing stage, followed by histidine, alanine and leucine. This result was slightly different from other studies on dry-cured hams, which reported that arginine, glutamic acid, leucine, lysine and alanine were the most amount free amino acids during processing [57].

Zhao et al. [58] found that the lower amino acids concentration of Jinhua ham was cysteine, which did not change very much during whole processing. This report was generally consistent with our research on dry-cured donkey leg. However, this result was inconsistent with the study on Laowo ham, which reported the amount of phenylalanine was lowest [14]. There was no significant increase in the total content of free amino acid from the green leg stage to the end of the salting stage (*p* > 0.05), which was due to the reduced activity of proteolytic enzymes in muscle caused by the lower ambient temperatures and increased salt content [59]. After the salting stage, the total amount of free amino acids increased significantly (*p* < 0.05). The reason for this result was the more suitable environmental conditions enhanced the activity of protease and then accelerated the rate of protein hydrolysis. In previous studies [57,60], the concentration of all free amino acids increased during dry-cured ham processing, which was the same as our research on dry-cured donkey leg.

The contents of most free amino acids in dry-cured donkey leg at the aging stage were about 10–16-fold of those in the green leg stage, which was much lower than that in Iberian dry-cured ham [61]. This was because our different samples and processing conditions. The relatively small increase in most free amino acids content was related to its possible involvement in Maillard reaction, which could make an important contribution to the characteristic flavor of dry-cured ham [62]. The most increased content of free amino acids in dry-cured donkey leg was arginine, followed by methionine, lysine and leucine.

All free amino acids detected in dry-cured donkey legs can be divided into four categories according to their sensory properties: umami amino acids (glutamic acid and aspartic acid), sweet amino acids (lysine and proline), bitter amino acids (methionine and histidine) and tasteless amino acids (lysine and proline), respectively [62]. As shown in Table 2, there was no significant increase in the four categories amino acids content from the green leg stage to the end of the salting stage of dry-cured donkey leg (*p* > 0.05). At the aging stage, the content of umami amino acids, sweet amino acids, bitter amino acids and tasteless amino acids increased significantly (*p* < 0.05) to 10.32, 12.15, 19.99 and 5.63 mg/g, respectively, which was about 7–9 times higher than those at the end of the salting stage. We also found that the concentration of bitter amino acids was the highest and increased greatly during processing, which might play an important role in forming volatile flavor compounds of dry-cured donkey leg.

### 3.3. Changes in Free Fatty Acids

The changes in free fatty acids can reflect the degree of lipid oxidation during processing of dry-cured ham and is also related to the changes of volatile flavor substances. As shown in Table 3, fifteen different fatty acids were found in dry-cured donkey leg at different processing stages, which were tested by the chromatography technique. The content of palmitic acid (C16:0) increased significantly from 3.43 mg/g at the green leg stage to 10.51 mg/g at the aging stage, which was considered the most abundant fatty acid at different stages of the processing. In addition, stearic acid (C18:0), oleic acid (C18:1) and linoleic acid (C18:2) were also present in relatively large proportions of the free fatty acids found in dry-cured donkey leg throughout processing. These fatty acid profiles are in agreement to those reported both in Laowo ham [14] and Yunnan dry-cured beef [63]. However, it was remarkably higher than the high oleic acid (C18:2) percentage reported both in dry-cured pig hams from Duroc and Mediterranean, followed by palmitic (C16:0) and stearic (C18:0) [20,60]. This difference could be possibly responsible for the typical sensorial characteristics of dry-cured donkey legs. Considering the concentrations of saturated fatty acids (SFA), monounsaturated fatty acids (MUFA) and polyunsaturated fatty acids (PUFA), they increased (*p* < 0.05) by 12.47, 9.67 and 14 mg/g from the green leg stage to the aging stage, respectively. This showed that free fatty acids were formed by the continuous decomposition of total fatty acids during processing. The amount of SFA in dry-cured donkey leg was the highest at each processing stage followed by PUFA and MUFA. This result does not agree with those found in Jinhua ham [28] and Laowo ham [14], where the PUFA and MUFA were the highest content, respectively. Moreover, it is possible that the differences may be a consequence of the animal species used. Miller et al. [64] suggested that fat composition of pigs was affected by their diet, and fatty acids from the foods accumulate in the fat of animals with little change. From the green leg stage to the end of the fermentation stage, the content of PFA increased by 3.9 mg/g, which is much lower than that of SFA and MFA. This may be due to the fact that PUFA is more susceptible to oxidative decomposition compared with SFA and MFA during curing, resulting in a greater decrease in this kind of fatty acids. It has been reported that PUFA of FFA fractions decreased during the process of dry-cured ham, which may be due to their high susceptibility to oxidation, whereas SFA and MUFA remained stable or increased [65].

The total content of free fatty acid increased significantly from the green leg to the end of the salting stage, mainly due to the increased activity of lipolytic enzymes, which led the hydrolysis of fats into fatty acids to become dominant. After salting, the total amount of free fatty acid increased insignificantly. The high salt content inhibited the lipase activity after the salting stage, which reduced the fat oxidation; moreover, unsaturated fatty acids (MUFA and PUFA) were gradually oxidized and degraded into characteristic volatile flavor substances of dry-cured donkey leg [32].

### 3.4. Volatile Flavor Compounds Analysis

Most of the volatile compounds of dry-cured ham are the result of complex biochemical reactions with the participation of dozens of muscle enzymes, including oxidation of unsaturated fatty acids and the interactions among proteins and peptides [4]. Moreover, some volatile compounds are generated by Strecker degradation of free amino acids and Maillard reactions [66]. Finally, the nonvolatile and volatile compounds play an important role in the development of the distinctive flavor of dry-cured ham [31]. As shown in Table 4, a total of 114 volatile flavor compounds were identified in dry-cured donkey leg by SPME-GC-MS, which is a slightly higher number than the 100 compounds identified in Serrano ham [67], similar to the 115 compounds found in dry-cured Laowo ham [14] and lower than the 182 volatile compounds reported in Chinese dry-cured hams (Mianning, Nuodeng, Saba and Sanchuan) [68].

About 30 compounds were detected at the green leg stage (0 d), 32 compounds were detected at the early salting stage (10 d), 41 compounds were detected at the middle salting stage (20 d), 49 compounds were detected at the end of salting (35 d), 62 compounds were detected at the early fermentation stage (65 d), 66 compounds were detected at the end of fermentation stage (105 d), 58 compounds were detected at the aging stage (165 d) and 28 compounds were increased by processing, respectively. All the detected volatile flavor compounds were categorized to 10 groups: 23 esters, 20 aldehydes, 20 hydrocarbons, 11 alcohols, 15 acids, 11 ketones, 6 pyrazines, 1 furans and 7 other compounds.

Results shown in Figure 1 indicate that the relative contents of acids, alcohols, aldehydes, phenols and pyrazines compounds showed an upward trend during the whole processing stage of dry-cured donkey leg, while the relative contents of hydrocarbons, esters, alcohols and ketones compounds showed a downward trend. Among the identified volatile compounds in the final products of dry-cured donkey leg, aldehydes, esters, alkane and alcohols were more abundant, with relative concentrations of 41.88%, 5.72%, 5.35% and 5.25%, respectively. This result was not agreement with Wu’s study, who found that hydrocarbons were the most abundant volatile compounds at the level of 23% in the final products of Laowo hams [14]. Some authors reported that aldehydes were the most important flavor compounds in the dry-cured hams, which is consistent with our research [69]. In comparison with the aging stage, about 24 volatile compounds were not found at the green leg stage. Some were aldehydes, such as octyl aldehyde, 2-phenyl-2-butenal, 3-ethylbenzaldehyde, trans-2-nonenal and trans-2-Heptenal; others were five alkanes, six esters, four alcohols, seven acids, two ketones, one furans, one pyrazines and one other compound. Yanjun et al. [16] reported that all the pyrazines and sulfur compounds detected at the postaging stage of Chinese Jinhua ham were not found at the green ham stage, which was different from our present study.

The different kinds of volatile compounds identified in dry-cured donkey leg changed a lot during different processing stages (*p* < 0.05). The aldehydes family was the first group of compounds, which was also considered as the major contributor to the flavor formation of dry-cured donkey leg. Aldehydes compounds reached their highest value (42.36%) at the end of the fermentation stage, and this chemical family showed insignificant change after the end of the salting stage (*p* > 0.05). Hexanal, octanal and nonanal compounds were the most abundant aldehydes during processing stages. Those aldehydes could be produced by oxidation of unsaturated fatty acids or Strecker reactions of some amino acids such as phenylalanine, isoleucine and leucine [22]. The amount of hexanal and nonanal compounds increased significantly during processing stages (*p* < 0.05), which might be due to the similar high relative humidity and temperature, indicating that the extent of lipid oxidation and may accelerate the generation of linoleic acid oxidation products [70]. Octanal deriving from oleic acid could provide a meat-like, green, fresh aroma or grass-like fruity notes, while (E)-2-octenal might give hams leafy, pungent, fatty and fruity sensations [71].

Esters compounds were found at a high proportion in dry-cured donkey leg, and they varied greatly during processing (*p* < 0.05). This chemical family was formed from the esterification of various acids and alcohols, which has little influence on the flavor of ham due to its high threshold value. Esters with long-chain acids have a fat odor, while esters with short-chain acids such as acetates, propanoates and butanoates give off fruity notes [71]. In the present study, the relative content of esters compounds significantly decreased from 11.58% at the green leg stage to 5.72% at the aging stage during processing of dry-cured donkey leg (*p* < 0.05). Our result was not agreement with Guo’s study, who found that the percentage of esters compounds increased by 1.96% from the green ham stage to the final product phase of mutton ham [13]. These differences could be explained by the changes in esterase during the processing stages.

Hydrocarbons can be formed via oxidative degradation of lipids, and they contribute little to the development of the dry-cured ham flavor because of the higher odor thresholds [31]. They accounted for 39.93% of the total volatile compounds at the green ham stage, which were much more than other chemical groups. In this study, n-alkanes (e.g., tetradecane, pentadecane, cetane), olefins (e.g., styrene) and their branched derivatives (e.g., 1,3-hexadiene, 2-methyltridecane) were mainly the compounds of the 20 hydrocarbons found in the processing of dry-cured donkey leg. Pentane was the most abundant hydrocarbon compound detected in three groups of dry-cured pig hams; however, it was not present in our study. Some aromatic chemicals, such as 1,3-m-xylene, 1,4-di-tert-butylbenzene, naphthalene and 2-methylnaphthalene were also found in this study, which were different from those identified in Chinese dry-cured hams [68]. This result could be related to the formation of special flavor in dry-cured donkey leg.

A total of 10 alcohol compounds were identified in dry-cured donkey leg, and the amount of them varied significantly in different stages during processing (*p* < 0.05). Alcohols are the key factors to the flavor of meat products due to their lower odor threshold, giving dry-cured hams herbaceous, woody and fatty notes [71]. For the identified alcohols, 1-hexanol was in the majority, followed by 1-octene-3-ol, heptanol and n-octanol. Heptanol and n-octanol were presented at each stage during processing, unlike other alcohols, which were not present at some stages. The oxidation of lipid or further reactions of their oxidation products could explain the presentation of lineal alcohols in dry-cured ham, while branched chain alcohols are most likely to arise from Streaker degradation of some amino acids or microbial fermentation [69]. Muriel et al. [72] found that the alcohols family had the higher proportion (at least 35%) in the final product of Iberian dry-cured ham, which is six times more than that in present study.

The acids family had the high threshold value and also played an important role in the special flavor of dry-cured donkey leg. This flavor substance has been identified as the product of microbial metabolism or produced by lipid oxidation reaction [72]. In this study, the contents of acids increased significantly from 1.99% at the green leg stage to 2.74% in the final product (*p* < 0.05). The changes in acids of dry-cured donkey leg were consistent with that in dry-cured Chinese Laowo ham [14] but opposite to that of dry-cured mutton ham [13] and Jinhua ham [73]. Of the 15 acids detected, caproic acid was the most abundant in the final product, followed by butyric acid and acetic acid.

Eleven different ketones were detected of dry-cured donkey leg during processing, whereas only one ketone was found in the green ham stage and the final product. Ketones, especially 2-ketones, were considered to give meat products characteristic aromas, such as ethereal, butter, spicy or blue cheese notes [74]. The content of ketones reached the highest value of 1.52% at the end of the salting stage and then decreased in later stages, showing that some ketones were converted to into carboxylic acids and other volatile flavor compounds during the fermenting and ripening stages. Among the detected ketones, 2,3-Octanedione and 2,5-Octanedione showed the highest percentages, which were 1.29%, 1.2%, respectively. Methyl-ketones such as 6-methyl-3,5-heptadien-2-one and 6-methyl-5-hepten-2-one have been associated with β-oxidation activity of mounds growing in the surface of dry-cured products.

Six pyrazines were found in this study, and they were not present from the green leg stage to the end of the salting stage. These detected pyrazines were mainly methylpyrazines including 2,5-dimethylpyrazine, 2,6-dimethylpyrazine, trimethylpyrazine, 2,3,5-trimethylpyrazine and tetramethylpyrazine. While pyrazines accounted for a small proportion of the identified flavor compounds, they could also give dry-cured hams a corn, nut, grilled, pungent and chocolate odor [71].

Furans provide a lot of vegetables-like aromatic notes to meat products due to their low odor thresholds [71]. Only one furan compound was identified in this study, which was 2-pentylfuran. 2-pentylfuran could be produced by the oxidation of linoleic acid and other n-6 fatty acids which was also found in Iberian dry-cured hams [75] and dry-cured mutton ham [13]. The contents of the other six flavor compounds, such as methoxyphenoxime, m-dimethyl ether and oleanthrone, were the lowest, making little effect on the overall flavor of dry-cured donkey leg.

## 4. Conclusions

The physical–chemical parameters, free fatty acids and free amino acids of dry-cured donkey legs were significantly affected by the processing stages. The accumulations of free amino acids and lipid oxidation have a great relationship with the formation of typical volatile flavor compounds of dry-cured donkey legs during processing time. The results obtained in this study help to more accurately understand the quality changes of dry-cured donkey leg during processing and provide a basis for improving product quality and optimizing the processing conditions in the future. Further studies are necessary to evaluate the sensory and microbiological aspects of the dry-cured donkey legs due to the high salt content of this meat product in present research.

## Figures and Tables

**Figure 1 foods-11-03542-f001:**
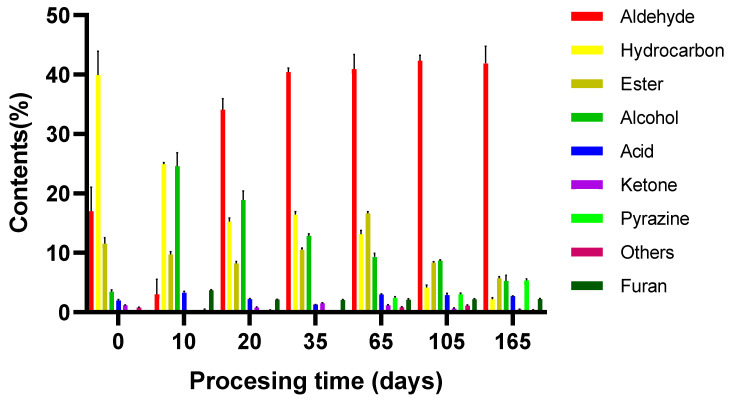
The relative contents of different flavor compounds detected in samples during the processing of dry-cured donkey leg. Processing times: 0 days: green leg; 10 days: start of salting; 20 days: middle of salting; 35 days: end of salting; 65 days: start of fermentation; 105 days: end of fermentation; 165 days: aging. The black bars of different lengths indicate significant differences between each group. (*p* < 0.05).

**Table 1 foods-11-03542-t001:** Physicochemical parameters during the processing of dry-cured donkey leg.

Physicochemical Parameters	Green Leg (0 d)	Start of Salting (10 d)	Middle of Salting (20 d)	End of Salting (35 d)	Start of Fermentation (65 d)	End of Fermentation (105 d)	Aging (165 d)
Moisture (%)	72 ± 0.03 ^a^	66 ± 0.02 ^b^	51 ± 0.02 ^c^	47 ± 0.01 ^d^	44 ± 0.02 ^d^	37 ± 0.01 ^e^	35 ± 0.01 ^e^
Water activity	0.90 ± 0.00 ^a^	0.90 ± 0.00 ^a^	0.89 ± 0.01 ^a^	0.87 ± 0.01 ^b^	0.84 ± 0.00 ^c^	0.80 ± 0.00 ^d^	0.78 ± 0.01 ^e^
pH	6.5 ± 0.03 ^a^	6.32 ± 0.05 ^bc^	6.28 ± 0.03 ^bc^	6.24 ± 0.05 ^c^	6.38 ± 0.04 ^b^	6.56 ± 0.11 ^a^	6.6 ± 0.07 ^a^
Chloride (% of DM)	0.45 ± 0.01 ^g^	1.36 ± 0.01 ^f^	3.16 ± 0.02 ^e^	6.46 ± 0.04 ^d^	9.87 ± 0.10 ^c^	10.89 ± 0.04 ^b^	12.39 ± 0.03 ^a^
TBARS (mg MDA/kg meat)	0.03 ± 0.00 ^f^	0.12 ± 0.01 ^e^	0.22 ± 0.01 ^d^	0.39 ± 0.01 ^a^	0.36 ± 0.02 ^ab^	0.33 ± 0.02 ^c^	0.34 ± 0.01 ^bc^
POV (g/100 g)	1.39 ± 0.01 ^e^	1.71 ± 0.03 ^d^	2.30 ± 0.09 ^c^	4.78 ± 0.03 ^b^	5.12 ± 0.21 ^a^	5.23 ± 0.01 ^a^	5.26 ± 0.01 ^a^
TVB-N (mg/kg)	1.43 ± 0.08 ^g^	2.00 ± 0.57 ^f^	3.34 ± 0.02 ^e^	3.72 ± 0.10 ^d^	5.59 ± 0.16 ^c^	6.74 ± 0.02 ^b^	8.58 ± 0.02 ^a^
Ash (%)	3 ± 0.00 ^e^	9 ± 0.00 ^d^	11 ± 0.01 ^c^	13 ± 0.00 ^b^	16 ± 0.01 ^a^	16 ± 0.00 ^a^	17 ± 0.00 ^a^

Note: Different letters (a–g) indicate a statistically significant difference within the same row (*p* < 0.05). Results expressed as means standard deviation (*n* = 6). Abbreviations: DM, dry matter; POV, peroxide; TBARS, thiobarbituric acid; TVB-N, total volatile basic nitrogen.

**Table 2 foods-11-03542-t002:** Changes of free amino acids in samples during the processing of dry-cured donkey leg (%).

	Green Leg (0 d)	Start of Salting (10 d)	Middle of Salting (20 d)	End of Salting (35 d)	Start of Fermentation (65 d)	End of Fermentation (105 d)	Aging (165 d)
Glutamate (Glu)	0.85 ± 0.04 ^d^	0.92 ± 0.05 ^d^	0.99 ± 0.01 ^d^	1.11 ± 0.10 ^d^	2.61 ± 0.10 ^c^	6.51 ± 0.29 ^b^	8.64 ± 0.15 ^a^
Aspartate (Asp)	0.20 ± 0.02 ^c^	0.22 ± 0.02 ^c^	0.25 ± 0.02 ^c^	0.27 ± 0.02 ^c^	0.60 ± 0.15 ^c^	1.03 ± 0.17 ^b^	1.68 ± 0.51 ^a^
Umami AAS	1.05 ± 0.06 ^d^	1.15 ± 0.04 ^d^	1.24 ± 0.01 ^d^	1.38 ± 0.12 ^d^	3.21 ± 0.15 ^c^	7.54 ± 0.40 ^b^	10.32 ± 0.36 ^a^
Alanine (Ala)	0.37 ± 0.03 ^d^	0.40 ± 0.02 ^d^	0.47 ± 0.02 ^d^	0.61 ± 0.03 ^d^	1.75 ± 0.43 ^c^	4.50 ± 0.39 ^b^	6.57 ± 0.44 ^a^
Glycine (Gly)	0.19 ± 0.03 ^d^	0.23 ± 0.01 ^d^	0.25 ± 0.04 ^d^	0.26 ± 0.02 ^d^	0.60 ± 0.07 ^c^	1.74 ± 0.05 ^b^	2.16 ± 0.05 ^a^
Serine (Ser)	0.20 ± 0.02 ^d^	0.24 ± 0.01 ^d^	0.24 ± 0.02 ^d^	0.27 ± 0.02 ^d^	0.66 ± 0.07 ^c^	1.80 ± 0.05 ^b^	2.26 ± 0.13 ^a^
Threonine (Thr)	0.14 ± 0.01 ^d^	0.15 ± 0.02 ^d^	0.16 ± 0.02 ^d^	0.18 ± 0.03 ^d^	0.55 ± 0.13 ^c^	0.74 ± 0.14 ^b^	1.16 ± 0.11 ^a^
Sweet AAs	0.90 ± 0.05 ^d^	1.03 ± 0.05 ^d^	1.13 ± 0.07 ^d^	1.32 ± 0.06 ^d^	3.55 ± 0.60 ^c^	8.78 ± 0.47 ^b^	12.15 ± 0.45 ^a^
Arginine (Arg)	0.11 ± 0.02 ^d^	0.14 ± 0.03 ^d^	0.18 ± 0.02 ^d^	0.22 ± 0.01 ^d^	0.39 ± 0.13 ^cd^	0.62 ± 0.07 ^b^	1.02 ± 0.19 ^a^
Isoleucine (Ile)	0.12 ± 0.02 ^d^	0.14 ± 0.03 ^d^	0.16 ± 0.02 ^d^	0.19 ± 0.02 ^d^	0.36 ± 0.07 ^c^	0.66 ± 0.11 ^b^	1.12 ± 0.11 ^a^
Leucine (Leu)	0.36 ± 0.03 ^d^	0.40 ± 0.02 ^d^	0.45 ± 0.06 ^d^	0.46 ± 0.04 ^d^	1.63 ± 0.11 ^c^	3.00 ± 0.30 ^b^	5.09 ± 0.19 ^a^
Methionine (Met)	0.06 ± 0.03 ^d^	0.06 ± 0.03 ^d^	0.07 ± 0.01 ^d^	0.12 ± 0.06 ^d^	0.22 ± 0.05 ^c^	0.55 ± 0.06 ^b^	1.02 ± 0.06 ^a^
Phenylalanine (Phe)	0.22 ± 0.03 ^d^	0.25 ± 0.02 ^d^	0.28 ± 0.02 ^d^	0.29 ± 0.02 ^d^	0.53 ± 0.09 ^c^	0.68 ± 0.11 ^b^	1.25 ± 0.09 ^a^
Tyrosine (Tyr)	0.14 ± 0.02 ^c^	0.17 ± 0.01 ^c^	0.22 ± 0.03 ^c^	0.24 ± 0.03 ^c^	0.44 ± 0.15 ^b^	0.56 ± 0.13 ^b^	1.18 ± 0.15 ^a^
Valine (Val)	0.12 ± 0.02 ^c^	0.16 ± 0.01 ^c^	0.24 ± 0.05 ^c^	0.25 ± 0.05 ^c^	0.52 ± 0.08 ^b^	0.67 ± 0.21 ^b^	1.25 ± 0.22 ^a^
Histidine (His)	0.55 ± 0.02 ^d^	0.60 ± 0.02 ^d^	0.66 ± 0.02 ^d^	0.75 ± 0.05 ^d^	2.03 ± 0.40 ^c^	5.66 ± 0.22 ^b^	7.85 ± 0.05 ^a^
Tryptophan (Try)	0.05 ± 0.01 ^d^	0.05 ± 0.02 ^d^	0.04 ± 0.01 ^d^	0.06 ± 0.02 ^d^	0.09 ± 0.02 ^cd^	0.16 ± 0.03 ^b^	0.22 ± 0.03 ^a^
Bitter AAs	1.73 ± 0.13 ^e^	1.98 ± 0.09 ^de^	2.30 ± 0.05 ^de^	2.59 ± 0.24 ^d^	6.21 ± 0.82 ^c^	12.57 ± 0.44 ^b^	19.99 ± 0.38 ^a^
Lysine (Lys)	0.29 ± 0.01 ^d^	0.33 ± 0.03 ^d^	0.37 ± 0.06 ^d^	0.40 ± 0.08 ^d^	1.60 ± 0.07 ^c^	2.47 ± 0.10 ^b^	4.69 ± 0.25 ^a^
Proline (Pro)	0.11 ± 0.02 ^c^	0.14 ± 0.02 ^c^	0.14 ± 0.05 ^c^	0.16 ± 0.02 ^c^	0.63 ± 0.10 ^b^	0.70 ± 0.04 ^b^	0.94 ± 0.08 ^a^
Tasteless AAs	0.40 ± 0.03 ^d^	0.46 ± 0.05 ^d^	0.51 ± 0.03 ^d^	0.56 ± 0.09 ^d^	2.23 ± 0.07 ^c^	3.17 ± 0.13 ^b^	5.63 ± 0.31 ^a^
Other Cysteine (Cys)	0.02 ± 0.01 ^c^	0.04 ± 0.01 ^c^	0.02 ± 0.02 ^c^	0.04 ± 0.01 ^c^	0.05 ± 0.03 ^c^	0.11 ± 0.02 ^b^	0.29 ± 0.02 ^a^
Total	4.10 ± 0.27 ^e^	4.66 ± 0.23 ^de^	5.20 ± 0.10 ^de^	5.89 ± 0.44 ^d^	15.26 ± 1.40 ^c^	32.18 ± 0.92 ^b^	48.37 ± 0.89 ^a^

Note: Different letters (a–e) indicate a statistically significant difference within the same row (*p* < 0.05). Results expressed as means standard deviation (*n* = 6).

**Table 3 foods-11-03542-t003:** Changes of free fatty acids in samples during the processing of dry-cured donkey leg (mg/g).

Free Fatty Acids	Green Leg (0 d)	Start of Salting (10 d)	Middle of Salting (20 d)	End of Salting (35 d)	Start of Fermentation (65 d)	End of Fermentation (105 d)	Aging (165 d)
C4:0	0.37 ± 0.07 ^ab^	0.27 ± 0.04 ^bcde^	0.26 ± 0.05 ^ae^	0.31 ± 0.09 ^ac^	0.41 ± 0.13 ^a^	0.25 ± 0.03 ^bcde^	0.29 ± 0.03 ^ad^
C14:0	0.00 ± 0.00 ^d^	0.07 ± 0.02 ^d^	0.36 ± 0.12 ^bc^	0.49 ± 0.27 ^ab^	0.20 ± 0.01 ^cd^	0.50 ± 0.01 ^b^	0.78 ± 0.20 ^a^
C16:0	3.43 ± 0.94 ^d^	3.83 ± 0.55 ^d^	5.76 ± 1.26 ^c^	6.24 ± 0.36 ^c^	8.49 ± 0.92 ^b^	8.40 ± 0.81 ^b^	10.51 ± 0.91 ^a^
C18:0	2.11 ± 0.92 ^e^	2.50 ± 0.12 ^e^	4.43 ± 0.28 ^bc^	3.00 ± 0.73 ^de^	5.07 ± 0.36 ^b^	3.92 ± 0. 79 ^cd^	6.91 ± 0.50 ^a^
C20:0	0.00 ± 0.00 ^b^	0.00 ± 0.00 ^b^	0.69 ± 0.58 ^a^	0.00 ± 0.00 ^b^	0.14 ± 0.03 ^b^	0.00 ± 0.00 ^b^	0.38 ± 0.14 ^ab^
SFAs	6.12 ± 1.58 ^e^	6.67 ± 0.81 ^e^	10.81 ± 1.2 ^cd^	10.04 ± 0.96 ^d^	14.32 ± 1.28 ^b^	13.08 ± 0.75 ^bc^	18.86 ± 1.23 ^a^
C16:1	0.00 ± 0.00 ^b^	0.32 ± 0.02 ^ab^	0.42 ± 0.06 ^a^	0.49 ± 0.12 ^a^	0.41 ± 0.14 ^a^	5.25 ± 0.34 ^a^	1.31 ± 0.21 ^a^
C18:1n 9c	0.21 ± 0.02 ^c^	0.00 ± 0.00 ^c^	1.51 ± 0.11 ^b^	0.27 ± 0.11 ^c^	1.19 ± 0.09 ^b^	0.63 ± 0.30 ^a^	5.25 ± 0.24 ^a^
C18:1n 9t	0.00 ± 0.00 ^c^	0.00 ± 0.00 ^c^	0.00 ± 0.00 ^c^	0.00 ± 0.00 ^c^	0.82 ± 0.39 ^b^	0.00 ± 0.00 ^c^	1.58 ± 0.23 ^a^
C20:1	0.00 ± 0.00 ^c^	0.00 ± 0.00 ^c^	0.00 ± 0.00 ^c^	0.00 ± 0.00 ^c^	0.70 ± 0.27 ^b^	0.00 ± 0.00 ^c^	1.75 ± 0.21 ^a^
MUFAs	0.21 ± 0.02 ^e^	0.32 ± 0.02 ^e^	1.93 ± 0.13 ^d^	0.76 ± 0.19 ^e^	3.12 ± 0.80 ^c^	5.88 ± 0.32 ^b^	9.89 ± 0.58 ^a^
C18:2n 6c	0.40 ± 0.28 ^d^	0.60 ± 0.05 ^d^	1.35 ± 028 ^c^	3.16 ± 0.20 ^b^	2.80 ± 0.48 ^b^	3.39 ± 0.06 ^b^	10.43 ± 0.60 ^a^
C18:2n6t	0.00 ± 0.00 ^c^	0.00 ± 0.00 ^c^	0.00 ± 0.00 ^c^	0.00 ± 0.00 ^c^	0.22 ± 0.10 ^b^	0.28 ± 0.25 ^b^	0.62 ± 0.02 ^a^
C18:3n3	0.19 ± 0.12 ^bcde^	0.29 ± 0.11 ^ac^	0.51 ± 0.07 ^a^	0.27 ± 0.02 ^bcd^	0.28 ± 0.13 ^ad^	0.21 ± 0.02 ^e^	0.41 ± 0.08 ^ab^
C18:3n6	0.00 ± 0.00 ^b^	0.00 ± 0.00 ^b^	0.00 ± 0.00 ^b^	0.64 ± 0.15 ^a^	0.00 ± 0.00 ^b^	0.00 ± 0.00 ^b^	0.32 ± 0.08 ^ab^
C20:2	0.00 ± 0.00 ^b^	0.00 ± 0.00 ^b^	0.00 ± 0.00 ^b^	0.00 ± 0.00 ^b^	0.00 ± 0.00 ^b^	0.00 ± 0.00 ^b^	0.34 ± 0.05 ^a^
C20:3	0.00 ± 0.00 ^c^	0.17 ± 0.03 ^c^	0.00 ± 0.00 ^d^	0.00 ± 0.00 ^d^	0.88 ± 0.16 ^b^	0.42 ± 0.26 ^b^	3.10 ± 0.48 ^a^
PUFAs	0.59 ± 0.40 ^d^	1.07 ± 0.08 ^cd^	1.86 ± 0.45 ^c^	4.06 ± 0.33 ^b^	4.17 ± 0.69 ^b^	4.30 ± 0.33 ^b^	15.52 ± 1.12 ^a^
Total	6.71 ± 1.25 ^d^	8.05 ± 0.61 ^d^	15.29 ± 0.8 ^c^	14.86 ± 1.19 ^c^	21.61 ± 2.70 ^b^	23.25 ± 1.42 ^b^	44.27 ± 2.70 ^a^

Note: Different letters (a–e) indicate a statistically significant difference within the same row (*p* < 0.05). Results expressed as means standard deviation (*n* = 6).

**Table 4 foods-11-03542-t004:** Changes of volatile flavor compounds in samples during the processing of dry-cured donkey leg (%).

Volatile Flavor Compounds	Green Leg (0 d)	Start of Salting (10 d)	Middle of Salting (20 d)	End of Salting (35 d)	Start of Fermentation (65 d)	End of Fermentation (105 d)	Aging (165 d)
Aldehydes	17.00 ± 4.07 ^c^	30.53 ± 2.56 ^b^	34.12 ± 1.82 ^b^	40.42 ± 0.7 ^a^	40.95 ± 2.43 ^a^	42.36 ± 0.89 ^a^	41.88 ± 2.89 ^a^
Hexanal	12.55 ± 3.15 ^d^	22.0 ± 2.14 ^bc^	22.6 ± 1.47 ^b^	18.44 ± 2.06 ^c^	27.18 ± 1.94 ^a^	28.38 ± 1.52 ^a^	27.27 ± 1.07 ^a^
N-octanal	ND	1.79 ± 0.52 ^c^	ND	4.36 ± 0.44 ^a^	ND	3.09 ± 0.37 ^b^	3.45 ± 0.45 ^b^
N-nonanal	3.48 ± 0.41 ^cde^	2.68 ± 1.02 ^de^	5.07 ± 1.32 ^b^	8.45 ± 1.09 ^a^	2.00 ± 0.30 ^e^	4.98 ± 0.21 ^bc^	3.97 ± 0.12 ^bcd^
2-phenyl-2-butenal	ND	ND	0.74 ± 0.25 ^a^	ND	ND	ND	0.03 ± 0.04 ^b^
Trans-2-Octenal	ND	1.55 ± 0.48 ^c^	2.19 ± 0.15 ^b^	1.67 ± 0.22 ^c^	2.91 ± 0.18 ^a^	ND	ND
Benzaldehyde	0.77 ± 0.43 ^c^	0.49 ± 0.14 ^c^	1.56 ± 0.17 ^b^	1.52 ± 0.11 ^b^	1.54 ± 0.05 ^b^	1.4 ± 0.16 ^b^	2.52 ± 0.19 ^a^
3-Ethylbenzaldehyde	ND	0.13 ± 0.03 ^bc^	0.29 ± 0.05 ^a^	0.29 ± 0.08 ^a^	ND	0.20 ± 0.02 ^b^	0.11 ± 0.02 ^c^
Trans-2-Nonenal	ND	ND	ND	1.33 ± 0.31 ^a^	1.44 ± 0.05 ^a^	0.7 ± 0.03 ^b^	0.92 ± 0.05 ^b^
2,4-Nonadienal	0.09 ± 0.11 ^c^	0.18 ± 0.03 ^c^	0.24 ± 0.08 ^bc^	0.42 ± 0.1 ^b^	1.02 ± 0.24 ^a^	0.18 ± 0.02 ^c^	0.2 ± 0.06 ^bc^
2,4-Decadienal	ND	0.25 ± 0.04 ^b^	0.07 ± 0.07 ^cd^	0.28 ± 0.02 ^b^	1.11 ± 0.14 ^a^	ND	0.16 ± 0.03 ^bc^
(E)-Tetradec-2-enal	ND	ND	ND	ND	ND	0.28 ± 0.01 ^b^	0.34 ± 0.04 ^a^
Trans,trans-2,4-Heptadienal	ND	ND	ND	0.13 ± 0.05 ^b^	0.13 ± 0.03 ^b^	0.15 ± 0.05 ^ab^	0.2 ± 0.02 ^a^
(E)-2-Decenal	ND	ND	ND	1.6 ± 0.14 ^a^	1.37 ± 0.1 ^b^	0.8 ± 0.02 ^c^	0.63 ± 0.02 ^d^
2-Dodecenal	ND	0 ^b^	ND	ND	ND	ND	0.24 ± 0.07 ^a^
Vanillin	0.11 ± 0.04 ^a^	ND	ND	0.05 ± 0.05 ^a^	0.05 ± 0.07 ^a^	ND	ND
Trans-2-Heptenal	ND	0.76 ± 0.04 ^cde^	0.85 ± 0.06 ^de^	0.97 ± 0.08 ^bcd^	1.7 ± 0.17 ^abc^	1.88 ± 0.13 ^a^	1.79 ± 1.13 ^ab^
Tetradecanal	ND	ND	ND	ND	ND	ND	0.05 ± 0.02 ^a^
Bourgeonal	ND	ND	0.49 ± 0.06 ^a^	ND	ND	ND	ND
2-Undecenal	ND	0.16 ± 0.02 ^cd^	ND	0.91 ± 0.17 ^a^	0.5 ± 0.13 ^b^	0.32 ± 0.1 ^bc^	ND
4-(1,1-dimethyl)Benzenepropanal	ND	0.48 ± 0.70 ^a^	ND	ND	ND	ND	ND
Hydrocarbons	39.93 ± 4.01 ^a^	24.94 ± 0.23 ^b^	15.29 ± 0.56 ^cd^	16.42 ± 0.51 ^c^	13.13 ± 0.62 ^d^	4.18 ± 0.37 ^e^	2.19 ± 0.26 ^e^
Octamethylcyclotetrasiloxane	17.53 ± 0.98 ^a^	6.63 ± 0.85 ^b^	5.35 ± 0.56 ^c^	4.4 ± 0.36 ^c^	4.53 ± 0.13 ^c^	ND	ND
Decamethylcyclopentasiloxane	17.65 ± 1.14 ^a^	13.13 ± 1.41 ^b^	5.16 ± 0.15 ^c^	6.47 ± 0.19 ^c^	ND	ND	ND
Dodecamethylcyclohexasiloxane	1.97 ± 0.79 ^b^	3.37 ± 0.51 ^a^	0.79 ± 0.32 ^c^	1.08 ± 0.1 ^c^	2.69 ± 0.16 ^ab^	0.58 ± 0.12 ^c^	0.42 ± 0.05 ^c^
Tetradecane	ND	0.72 ± 0.05 ^b^	1.5 ± 0.22 ^a^	1.61 ± 0.07 ^a^	0.62 ± 0.06 ^b^	0.59 ± 0.05 ^b^	0.53 ± 0.04 ^b^
Pentadecane	ND	0.9 ± 0.2 ^a^	0.57 ± 0.13 ^b^	0.44 ± 0.03 ^bc^	0.27 ± 0.06 ^c^	0.51 ± 0.04 ^b^	0.55 ± 0.12 ^b^
N-Hexadecane	ND	ND	ND	0.11 ± 0.03 ^a^	ND	ND	ND
Octadecane	ND	ND	ND	0.52 ± 0.07 ^a^	ND	ND	ND
1,4-di-tert-butylbenzene	0.46 ± 0.15 ^a^	ND	ND	ND	ND	ND	ND
Tetradecamethylcycloheptasiloxane	0.19 ± 0.1 ^bc^	0.18 ± 0.1 ^c^	0.23 ± 0.06 ^bc^	0.38 ± 0.15 ^ab^	0.53 ± 0.13 ^a^	0.15 ± 0.04 ^c^	0.12 ± 0.03 ^c^
Naphthalene	ND	ND	0.04 ± 0.04 ^ab^	0.03 ± 0.04 ^ab^	ND	ND	0.05 ± 0.03 ^a^
2-Methylnaphthalene	0.13 ± 0.05 ^a^	0.01 ± 0.01 ^c^	0.04 ± 0.04 ^bc^	0.07 ± 0.02 ^b^	0.13 ± 0.03 ^a^	ND	0.08 ± 0.01 ^ab^
1-Methylnaphthalene	ND	ND	0.08 ± 0.05 ^a^	ND	0.07 ± 0.05 ^a^	0.04 ± 0.03 ^ab^	0.03 ± 0.03 ^ab^
N-Tridecane	ND	ND	0.95 ± 0.14 ^a^	ND	0.59 ± 0.13 ^b^	ND	ND
2-Methyltridecane	ND	ND	ND	ND	0.24 ± 0.05 ^a^	ND	0.23 ± 0.04 ^a^
Dichloromethane	ND	ND	ND	ND	1.04 ± 0.06 ^a^	1.04 ± 0.06 ^a^	ND
1,3-Hexadiene	ND	ND	ND	ND	ND	0.3 ± 0.15 ^a^	ND
Styrene	ND	ND	ND	ND	2.42 ± 0.07 ^a^	ND	ND
Propyl cyclopropane	ND	ND	ND	ND	ND	0.64 ± 0.03 a	ND
Benzocycloheptatriene	ND	ND	ND	0.05 ± 0.02 ^b^	ND	ND	0.18 ± 0.03 ^a^
Esters	11.58 ± 1 ^b^	9.72 ± 0.47 ^c^	8.21 ± 0.37 ^d^	10.51 ± 0.29 ^c^	16.64 ± 0.3 ^a^	8.37 ± 0.16 ^d^	5.72 ± 0.25 ^e^
Methyl octanoate	1.44 ± 0.23 ^b^	1.46 ± 0.19 ^b^	1.2 ± 0.04 ^bc^	0.98 ± 0.12 ^c^	1.82 ± 0.06 ^a^	1.43 ± 0.09 ^b^	1.37 ± 0.11 ^b^
Methyl pelargonate	0.5 ± 0.11 ^c^	0.26 ± 0.05 ^d^	0.78 ± 0.09 ^b^	0.39 ± 0.12 ^cd^	1.35 ± 0.05 ^a^	0.85 ± 0.08 ^b^	0.55 ± 0.09 ^c^
Methyl decanoate	2.77 ± 0.53 ^a^	0.12 ± 0.04 ^b^	0.26 ± 0.08 ^b^	0.3 ± 0.05 ^b^	0.31 ± 0.11 ^b^	0.41 ± 0.06 ^b^	0.4 ± 0.11 ^b^
Ethyl decanoate	ND	ND	ND	ND	ND	ND	0.05 ± 0.02 a
Hexyl hexoate	ND	ND	ND	ND	0.18 ± 0.04 ^a^	0.18 ± 0.02 ^a^	ND
Ethyl caproate	ND	ND	ND	ND	0.2 ± 0.08 ^a^	0.22 ± 0.04 ^a^	ND
Amyl hexoate	ND	ND	ND	ND	ND	0.2 ± 0.05 ^a^	ND
Gamma-Butyrolactone	ND	ND	ND	ND	0.05 ± 0.03 ^a^	0.05 ± 0.04 ^a^	0.05 ± 0.01 ^a^
Gamma-Octanoiclacton	ND	ND	ND	ND	0.3 ± 0.09 ^a^	0.13 ± 0.04 ^b^	0.11 ± 0.04 ^b^
Methyl dodecanoate	0.16 ± 0.06 ^a^	ND	ND	ND	ND	ND	ND
Delta-Octalactone	6.15 ± 0.29 ^d^	7.88 ± 0.4 ^b^	5.97 ± 0.22 ^d^	7.17 ± 0.62 ^c^	9.75 ± 0.42 ^a^	2.33 ± 0.1 ^e^	2.09 ± 0.09 ^e^
Gamma-Nonalactone	0.15 ± 0.04 ^c^	ND	ND	ND	0.68 ± 0.04 ^a^	0.24 ± 0.06 ^b^	0.14 ± 0.04 ^c^
Methyl myristate	0.14 ± 0.02 ^a^	ND	ND	ND	ND	ND	ND
Methyl palmitate	0.17 ± 0.03 ^a^	ND	ND	ND	ND	ND	ND
Methyl palmitoleate	0.1 ± 0.03 ^a^	ND	ND	ND	ND	ND	ND
Delta-Dodecalactone	ND	ND	ND	ND	ND	ND	0.03 ± 0.03 ^a^
Methyl benzoate	ND	ND	ND	ND	0.07 ± 0.05 ^a^	0.07 ± 0.03 ^a^	ND
Methyl phenylacetate	ND	ND	ND	0.27 ± 0.07 ^b^	0.24 ± 0.04 ^b^	0.76 ± 0.11 ^a^	0.84 ± 0.14 ^a^
Ethyl phenylacetate	ND	ND	ND	ND	ND	0.06 ± 0.03 ^b^	0.09 ± 0.01 ^a^
Methyl 2-octenate	ND	ND	ND	ND	0.75 ± 0.1 ^a^	0.75 ± 0.01 ^a^	ND
Undecanolactone	ND	ND	ND	ND	0.29 ± 0.03 ^a^	0.04 ± 0.01 ^b^	ND
Butyl decyl ester	ND	ND	ND	0.15 ± 0.04 ^a^	ND	ND	ND
Alcohols	3.48 ± 0.3 ^e^	24.59 ± 2.26 ^a^	18.88 ± 1.5 ^b^	12.86 ± 0.31 ^c^	9.3 ± 0.62 ^d^	8.68 ± 0.12 ^d^	5.25 ± 0.96 ^e^
1-Octen-3-Ol	2.00 ± 0.08 ^e^	9.10 ± 0.32 ^a^	7.17 ± 0.22 ^b^	5.31 ± 0.09 ^c^	3.53 ± 0.37 ^d^	3.55 ± 0.08 ^d^	ND
Heptanol	0.80 ± 0.14 ^d^	0.85 ± 0.08 ^d^	1.39 ± 0.16 ^b^	1.66 ± 0.08 ^a^	0.97 ± 0.10 ^cd^	1.4 ± 0.07 ^b^	1.17 ± 0.03 ^c^
N-Octanol	0.68 ± 0.13 ^d^	0.7 ± 0.06 ^d^	1.73 ± 0.11 ^a^	1.85 ± 0.07 ^a^	1.35 ± 0.06 ^b^	1.3 ± 0.05 ^b^	1.02 ± 0.06 ^c^
1-Pentanol	ND	2.05 ± 0.17 ^a^	0 ± 0 d	1.53 ± 0.1 ^c^	1.61 ± 0.12 ^c^	1.84 ± 0.05 ^b^	0 ± 0 ^d^
N-Hexanol	ND	11.89 ± 1.89 ^a^	6.83 ± 0.09 ^b^	2.43 ± 0.29 ^c^	1.49 ± 0.05 ^cd^	0 ^d^	2.31 ± 1.02 ^c^
Benzyl alcohol	ND	ND	0.09 ± 0.03 ^a^	0.02 ± 0.03 ^b^	0.03 ± 0.03 ^b^	0.03 ± 0.03 ^b^	0.02 ± 0.03 ^b^
Phenethyl alcohol	ND	ND	0.25 ± 0.07 ^b^	0.06 ± 0.02 ^c^	0.32 ± 0.09 ^b^	0.52 ± 0.06 ^a^	0.32 ± 0.07 ^b^
trans-2-Octen-1-ol	ND	ND	ND	ND	ND	ND	0.41 ± 0.12 ^a^
Hexaethylene glycol	ND	ND	ND	ND	ND	0.04 ± 0.02 ^a^	ND
Epoxydihydrolinalool	ND	ND	0.36 ± 0.06 ^a^	ND	ND	ND	ND
Acids	1.99 ± 0.17 ^c^	3.32 ± 0.21 ^a^	2.21 ± 0.1 ^c^	1.3 ± 0.02 ^d^	2.98 ± 0.12 ^b^	2.9 ± 0.29 ^b^	2.74 ± 0.04 ^b^
Hexanoic acid	0.46 ± 0.08 ^b^	1.98 ± 0.13 ^a^	1.96 ± 0.08 ^a^	0.66 ± 0.03 ^b^	2.11 ± 0.15 ^a^	0.67 ± 0.14 ^b^	0.65 ± 0.08 ^b^
Acetic acid	ND	ND	ND	ND	ND	0.58 ± 0.05 ^a^	0.41 ± 0.06 ^b^
Dimethyldithiocarbamate	0.08 ± 0.03 ^a^	ND	ND	ND	ND	ND	ND
N-Hexadecanoic acid	1.2 ± 0.14 ^b^	1.34 ± 0.08 ^a^	ND	ND	ND	ND	ND
Butanoic acid	ND	ND	ND	0.28 ± 0.03 ^c^	0.17 ± 0.02 ^d^	0.74 ± 0.08 ^a^	0.5 ± 0.02 ^b^
Isobutyric acid	ND	ND	ND	ND	ND	0.24 ± 0.03 ^a^	0.25 ± 0.01 ^a^
Butanoic acid, 2-methyl-, 3-methy	ND	ND	ND	ND	ND	ND	0.34 ± 0.06 ^a^
Butanoic acid, 2-methyl-, hexyl	ND	ND	ND	ND	ND	ND	0.43 ± 0.04 ^a^
Octanoic acid	ND	ND	0.13 ± 0.02 ^bc^	0.08 ± 0.04 ^c^	0.23 ± 0.07 ^a^	0.14 ± 0.01 ^b^	ND
Nonoic acid	ND	ND	0.06 ± 0.04 ^b^	ND	0.13 ± 0.02 ^a^	0.05 ± 0.04 ^b^	ND
Heptanoic acid	ND	ND	0.06 ± 0.02 ^a^	ND	0.06 ± 0.03 ^a^	0.05 ± 0.02 ^a^	0.04 ± 0.02 ^a^
Decoic acid	0.25 ± 0.04 ^a^	ND	ND	ND	ND	ND	ND
Valeric acid	ND	ND	ND	ND	ND	0.28 ± 0.04 ^a^	0.12 ± 0.03 ^b^
Dodecanoic acid	ND	ND	ND	ND	ND	0.02 ± 0.03 ^a^	ND
2-methyl-propionic acid	ND	ND	ND	0.28 ± 0.06 ^a^	0.28 ± 0.04 ^a^	0.13 ± 0.02 ^b^	ND
Ketones	1.2 ± 0.02 ^b^	0.13 ± 0.05 ^f^	0.81 ± 0.1 ^c^	1.52 ± 0.07 ^a^	1.18 ± 0.11 ^b^	0.66 ± 0.08 ^d^	0.48 ± 0.04 ^e^
2(3H)-Furanone, 5-butyldihydro-	ND	ND	ND	0.02 ± 0.01 ^a^	ND	ND	ND
Methyl heptenone	ND	ND	0.37 ± 0.04 ^a^	ND	ND	ND	ND
6-Methyl-3,5-heptadiene-2-one	ND	ND	0.26 ± 0.03 ^a^	ND	ND	0.09 ± 0.03 ^b^	0.11 ± 0.04 ^b^
6-Methyl-5-heptadiene-2-one	ND	ND	ND	ND	0.16 ± 0.01 ^a^	ND	ND
Nerylacetone	ND	ND	0.18 ± 0.03 ^a^	0.14 ± 0.02 ^b^	ND	ND	ND
Geranylaceto	ND	0.13 ± 0.05 ^a^	ND	ND	0.1 ± 0.06 ^a^	0.1 ± 0.04 ^a^	ND
Acetophenone	ND	ND	ND	ND	0.09 ± 0.04 ^a^	0.09 ± 0.03 ^a^	ND
2-Piperidinone	ND	ND	ND	0.07 ± 0.01 ^a^	ND	ND	ND
2,3-Octanedione	ND	ND	ND	1.29 ± 0.07 ^a^	ND	ND	ND
2,5-Octanedione	1.2 ± 0.02 ^a^	ND	ND	ND	ND	ND	ND
3,5-Octadien-2-one	ND	ND	ND	ND	0.83 ± 0.04 ^a^	0.38 ± 0.03 ^b^	0.37 ± 0.05 ^b^
2,5-dimethylpyrazine	ND	ND	ND	ND	1.18 ± 0.04 ^b^	1.35 ± 0.07 ^a^	1.33 ± 0.09 ^a^
Pyrazines	ND	ND	ND	ND	2.44 ± 0.17 ^c^	3.03 ± 0.18 ^b^	5.35 ± 0.24 ^a^
2,6-dimethylpyrazine	ND	ND	ND	ND	ND	ND	0.77 ± 0.08 ^a^
Trimethyl-pyrazine	ND	ND	ND	ND	ND	ND	2.38 ± 0.24 ^a^
2,3,5-Trimethyl-6-ethylpyrazine	ND	ND	ND	ND	0.11 ± 0.01 ^b^	0.13 ± 0.05 ^b^	0.27 ± 0.05 ^a^
Trimethyl-pyrazine	ND	ND	ND	ND	0.88 ± 0.13 ^b^	1.3 ± 0.04 ^a^	ND
Tetramethylpyrazine	ND	ND	ND	ND	0.27 ± 0.06 ^b^	0.25 ± 0.07 ^b^	0.6 ± 0.06 ^a^
Furans	0 ^c^	3.68 ± 0.12 ^a^	2.13 ± 0.04 ^b^	2.06 ± 0.08 ^b^	2.1 ± 0.16 ^b^	2.18 ± 0.11 ^b^	2.23 ± 0.08 ^b^
2-Pentylfuran	0 ^c^	3.68 ± 0.12 ^a^	2.13 ± 0.04 ^b^	2.06 ± 0.08 ^b^	2.1 ± 0.16 ^b^	2.18 ± 0.11 ^b^	2.23 ± 0.08 ^b^
Others	0.8 ± 0.07 ^b^	0.4 ± 0.1 ^c^	0.36 ± 0.04 ^cd^	0.24 ± 0.03 ^d^	0.85 ± 0.08 ^b^	1.12 ± 0.12 ^a^	0.36 ± 0.08 ^cd^
Methoxy phenoxime	0.64 ± 0.06 ^a^	ND	ND	ND	0.53 ± 0.09 ^b^	0.68 ± 0.04 ^a^	ND
1H-Indene,1-methilene-	ND	ND	ND	ND	0.1 ± 0.03 ^a^	0.09 ± 0.04 ^a^	ND
1,3-dimethoxy-Benzene	ND	ND	ND	ND	0.22 ± 0.02 ^b^	0.35 ± 0.07 ^a^	ND
Sec-Butylamine	ND	0.4 ± 0.1 ^a^	ND	ND	ND	ND	ND
N,N-dibutyl-Formamide	ND	ND	0.36 ± 0.04 ^a^	0.24 ± 0.03 ^b^	ND	ND	ND
N-Hydroxymethylacetamide	ND	ND	ND	ND	ND	ND	0.36 ± 0.08 ^a^
Azulene	0.16 ± 0.03 ^a^	ND	ND	ND	ND	ND	ND

Note: Different letters (a–f) indicate a statistically significant difference within the same row (*p* < 0.05). Results expressed as means standard deviation (*n* = 6). ND indicates not detected.

## Data Availability

Data is contained within the article.

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
