# Peer review of "Changes in the Physical–Chemical Properties and Volatile Flavor Components of Dry-Cured Donkey Leg during Processing"

_foods, 2022, doi:10.3390/foods11213542_

Round 1

Reviewer 1 Report

Title: Changes in the physical-chemical properties and volatile flavor components of dry-cured donkey ham during processing

Journal: Foods

In general, the work provides new information on the physical-chemical characteristics of the cured donkey leg. Do you really eat donkey meat? How many kilos per person? Official data, please.

The term "ham" is not well used: “Ham” is the product made from the hind limb of the PIG subjected to a preservation by a process of curing. Therefore, authors should therefore substitute the term "ham" for "leg" throughout the document when referring to donkey, and only use ham when referring to pig.

Authors should explain the type of stunning used prior to slaughter.

The authors should justify the parameters (temperature and humidity) used in the experiment, since they are not justified:v The temperature used in the first stages of salting is surprising, excessively high (7ºC). During the first phases of salting, the hams must be at temperatures below 4º C, since the water activity in this first stage is still high. It is also surprising that the humidity applied for curing increases with the time of the process and would cause the piece to crust or rot.

They must specify the characteristics of the tube where the samples are stored at -80ºC.

The methods of section 2.3. must be explained in detail.

What are the reasons that try to explain the variation of the TBA (lines 263-268) based on?

Add the conditions of document 53 referenced on line 272.

In the conclusions section: a) the authors refer to the processing method factor. The work does not have the processing method as a factor studied, but rather the factor studied is "the curing phases". b) the authors should comment on the practical implications of the study and not repeat the results obtained and already commented on in the results section. This part should be corrected and rewritten.

I

Author Response

Dear reviewer,

Thanks for your constructive comments on our manuscript. We have carefully considered the suggestions and made some changes. Please see the attachment.

Have a nice day!

Reviewer 2 Report

This work characterizes the physicochemical profile of dry-cured donkey ham during processing. The results are well described and adequately discussed, but some details need to be improved before publication, which are shown below:

Materials and methods

Review the guide for authors on the use of units. In this section “mL” and “ml” have been used, which should be standardized. Also, check the separation of “°C” and “min” from the numbers.

Lines 118-124. Please, give more details of the physicochemical analyzes.

Line 151. Correct Floch by Folch.

Line 154. Indicate the temperature.

Line 174 and 185. Use µ instead of u to express µg or µL. See this observation in the whole manuscript.

Lines 191-192. Indicate how long the samples were thawed.

Lines 211-216. It is not clear what are the treatments or independent variables to justify the use of ANOVA, so the authors should specify it in section 2.1. Also, indicate the experimental design in the study.

 Results and discussion

Lines 264-264. Cite references in the explanation of those lines.

Line 294. Change Table 3 by Table 2.

Line 407. If the results are expressed with p-value, the bars in Figure 1 should show significant differences between times of each group of compounds.

Figure 1. Give a suitable title to the figure, not only expressing what the times mean, but also the meaning of the bars.

Conclusions

Lines 510-511. Processing method or processing time?

Author Response

(The authors gave the same response as above.)

Reviewer 3 Report

Dear authors, 

The investigation is novel as the changes in the physical-chemical properties and volatile flavor components of dry-cured donkey ham are evaluated taking into account 6 stages of processing.

The research presents results of interest. However, the following observations should be reviewed:

In general, the manuscript must be reviewed by a native English language speaker.

Abstract

Lines 23- 26:  This information must be restructured for a better understanding. It is recommended mentioning the figures first and then the variables determined.

Line 27: In which units are the TBARS reported?   

Lines 46-48: This section must be restructured for a better understanding. Use the connectors correctly.                   

Line 74: Change “though” to “through”

Materials and methods

In general, the methodology section is not well written which hinders understanding.

Line 93: Which breed was used for this investigation?       

Line 100: Use a period after “legs”. 

Line 101: Why is the exact amount of salt added not stated?        

Line 115: Place “and” before “end”.

Lines 119-121: If this is the case, place the word “respectively” after the last methodology.

Line 124: Mention in which units the TBARS are reported. 

Line 128: Do not use the definite article “The” before a figure. Correct this throughout the manuscript.

Line 152: Do no start a sentence with a figure.       

Line 154:  Separate the figures from the units. Apply this to the whole manuscript where necessary.

Results and discussion

For most of the results, the authors make comparisons with pork hams. Instead, it is recommended making comparisons with ruminant meat.

Line 222: The authors state …” and the highest decrease occurred at the middle of salting stage (p < 0.05).”… This does not make sense, what is meant by this?

Lines 224-226: I do not agree with the use of the word “slightly” as there is a considerable difference between the 35% and 54% figures. Mention in which animal species this percentage of humidity was reached. 

Lines 230-231: Is the way in which the results for the water activity are mentioned correct? Water activity is dimensionless.

Line 256: Specify which oxidation products.

Line 288: Mention what the national standard is.

Line 319: Place “and” after “lysine”.

Line 323- 325: This is unclear. Which are the 4 amino acids?

Line 327: Use the word “respectively” after mentioning the figures.

Lines 364-365:  Moreover, it is possible that the differences may be a consequence of the animal species used.

Line 378: It is suggested replacing “On the other hand” with another expression.

Table 1.

Review and correct the way in which the humidity results are displayed. The way they are mentioned in the manuscript is different to the way that they are mentioned in the table.

It is suggested that the results (figures) for Chloride (%) be rounded.

Figure 1. In order for the compounds to be better differentiated, it is suggested changing the color of the hydrocarbons bar. 

Author Response

(The authors gave the same response as above.)

Reviewer 4 Report

Dear Authors,

 After Changes in the physical-chemical properties and volatile flavor components of dry-cured donkey ham during processing

I was able to identify many details, such as the reduction of spaces between characters or the incorporation of these. The corrections can be easily incorporated. You will find them highlighted inside the PDF. Additionally, you will find some suggestions inside text balloons (inside the pdf file).

Congratulations

Author Response

Dear reviewer,

Thanks for all your good suggestions. I have corrected the manuscript according to your comments. Please see the attachment.

Have a nice day!

Round 2

Reviewer 1 Report

xThe authors have responded to the requested corrections. I now accept the publication of the study in Foods.

Author Response

Dear reviewer,

Thank you again for your comments on my manuscript!

Have a nice day!

Reviewer 3 Report

Dear authors,

Although the manuscript has been improved according to the suggestions, the following should be reviewed:

Firstly, it is requested that it is specifically mentioned in which lines the changes were made in response to the observations made by the reviewers.

 Line 27: Once again, mention must be made of the units used to express the TBARS.

 Lines 24-27: Move the word “respectively” to the end of the sentence.

 Lines 783: Replace “ctegories” with “categories”

Author Response

Dear reviewer,

Thank you again for your very careful comments on our manuscript!

We have carefully made some changes. Please see the attachment.

Have a nice day!
